# Safety of Immune Checkpoint Inhibitor Resumption after Interruption for Immune-Related Adverse Events, a Narrative Review

**DOI:** 10.3390/cancers14040955

**Published:** 2022-02-14

**Authors:** Marion Allouchery, Clément Beuvon, Marie-Christine Pérault-Pochat, Pascal Roblot, Mathieu Puyade, Mickaël Martin

**Affiliations:** 1Pharmacologie Clinique et Vigilances, Centre Hospitalier Universitaire de Poitiers, 2 Rue de la Milétrie, 86000 Poitiers, France; marion.allouchery@chu-poitiers.fr (M.A.); marie-christine.perault@chu-poitiers.fr (M.-C.P.-P.); 2Université de Poitiers, 15 Rue de l’Hôtel-Dieu, TSA 71117, 86000 Poitiers, France; clement.beuvon@chu-poitiers.fr (C.B.); pascal.roblot@chu-poitiers.fr (P.R.); 3Médecine Interne et Maladies Infectieuses, Centre Hospitalier Universitaire de Poitiers, 2 Rue de la Milétrie, 86000 Poitiers, France; mathieu.puyade@chu-poitiers.fr; 4CIC-1402, Centre Hospitalier Universitaire de Poitiers, 2 Rue de la Milétrie, 86000 Poitiers, France; 5Laboratoire de Neurosciences Expérimentales et Cliniques, INSERM U1084, Université de Poitiers, 1 Rue Georges Bonnet, 86073 Poitiers, France; 6INSERM U1313, Centre Hospitalier Universitaire de Poitiers, 2 Rue de la Milétrie, 86000 Poitiers, France

**Keywords:** safety, immune-related adverse events, immune checkpoint inhibitors

## Abstract

**Simple Summary:**

While immune checkpoint inhibitors (ICIs) have become the standard of care for several types of cancer, they are closely associated with specific immune-related adverse events (irAEs). The decision to resume ICI treatment after its interruption due to irAEs is challenged by the need for tumor control versus the risk of occurrence of the same or different irAEs. Data regarding the safety of ICI resumption after irAE remain scarce, heterogeneous, and mostly based on small samples of patients or focused only on the recurrence rate of the same irAE. Herein, we provided a narrative review on the safety of ICI resumption after interruption due to irAE(s).

**Abstract:**

Immune checkpoint inhibitors (ICIs) have become the standard of care for several types of cancer due to their superiority in terms of survival benefits in first- and second-line treatments compared to conventional therapies, and they present a better safety profile (lower absolute number of grade 1–5 adverse events), especially if used in monotherapy. However, the pattern of ICI-related adverse events is totally different, as they are characterized by the development of specific immune-related adverse events (irAEs) that are unique in terms of the organs involved, onset patterns, and severity. The decision to resume ICI treatment after its interruption due to irAEs is challenged by the need for tumor control versus the risk of occurrence of the same or different irAEs. Studies that specifically assess this point remain scarce, heterogenous and mostly based on small samples of patients or focused only on the recurrence rate of the same irAE after ICI resumption. Moreover, patients with grade ≥3 irAEs were excluded from many of these studies. Herein, we provide a narrative review on the field of safety of ICI resumption after interruption due to irAE(s).

## 1. Introduction

Immune checkpoint inhibitors (ICIs) are monoclonal antibodies directed against cytotoxic T-lymphocyte antigen 4 (CTLA-4) (ipilimumab), programmed cell death protein 1 (PD-1) (pembrolizumab, nivolumab, cemiplimab) or the PD-1 ligand (PDL-1/2) (atezolizumab, durvalumab, and avelumab). ICIs are the standard of care for several types of cancers due to their superiority in terms of survival benefits as first- and second-line treatments, compared to conventional therapies [1,2,3,4,5,6].

The engagement of CTLA-4 in T cells during the induction phase of an anti-tumor immune response impedes T cell activation by inhibition of the co-stimulation signal, leading to anergy. Furthermore, engagement of PD-1 on effector T cells with PDL-1 in tumor-associated antigen-presenting cells promotes T cell exhaustion. ICIs, by releasing these inhibitory brakes of T cells, promote anti-tumor T cell activation and the maintenance of anti-tumor T cell effector function [7]. ICIs also work by activating other cells of the innate and adaptive immune system, leading to a coordinated and effective anti-tumoral response [1].

Compared to conventional therapy, ICIs are associated with a better safety profile (lower absolute number of grade 1–5 adverse events), especially if used in monotherapy [8]. However, the pattern of ICI-related adverse events is quite different, with the development of specific immune-related adverse events (irAEs) that are unique in terms of organs involved, onset patterns, and severity [7,9]. The mechanisms of these irAEs are related to the non-specific activation of the adaptive immune system [1]. Indeed, anti-CTLA-4 and anti-PD(L)-1 are associated with the reduced survival and inhibitory functions of CD4^+^ CD25^+^ regulatory T cells (Treg), an increased proportion of type 17 T helper cells (anti-CTLA-4) and cytokine production, in addition to the induction of cross-reactivity between anti-tumor T cells and antigens on healthy cells, leading to autoantibody production, tissue injury and auto-immune diseases [7].

IrAEs are highly heterogenous in terms of time taken until occurrence [10,11,12], type [8] and severity [13], depending on the ICI regimen [14] and/or cancer type [13]. Moreover, individual variations of the same cancer and ICI combination suggest a genetic background. However, to date, the risk factors for irAEs remain unknown [15].

Several recommendations have been published to help to manage irAEs [16,17]. Globally, grade 2/3 irAEs require corticosteroids and temporary ICI discontinuation, with the possibility of resuming when symptoms revert to grade ≤1, although permanent ICI discontinuation is recommended for grade 4 irAEs (except for endocrinopathies if controlled by hormone replacement). The decision to resume ICI treatment after grade ≥2 irAEs is challenged by the need for tumor control versus the risk of occurrence of the same or different irAEs. Studies having specifically assessed the safety of ICI retreatment after ICI discontinuation for irAEs remain scarce and heterogenous. Most are based on small samples of patients [18,19,20] or focused only on the recurrence rate of the same irAE after an ICI rechallenge [21]. Moreover, patients with grade ≥3 irAEs have been excluded from many of these studies [22,23,24,25]. To better understand the safety profile of ICI resumption after irAE occurrence, we conducted a narrative literature review to summarize the available data on the safety of ICI resumption after discontinuation for irAEs.

## 2. When Time and Words Matter to Define the Resumption of ICI(s) and Second irAEs

The first pitfalls when comparing data are the timeline of ICI rechallenge and second irAEs onset. The absence of definitions in terms of a timeline may lead to heterogeneous definitions of second irAEs. The experience from “natural” auto-immune diseases suggests that the break of tolerance may sometimes occur many years before onset of the auto-immune disease [26]. Due to the rapid onset of irAEs (a few weeks after the beginning of the treatment), it could be hypothesized that the break of tolerance at a cellular level occurred before ICI treatment, but without any clinical symptoms of auto-immune disease. As a result, the stereotypical kinetics of the first irAEs (skin irAEs between 3–7 weeks, pulmonary irAEs between 10–16 weeks, hepatitis between 6–15 weeks, colitis between 4–10 weeks, and endocrinopathies after 6 weeks for anti-PD-(L)1) [27] may be related to the activation and amplification of the preexisting autoimmune cells, which may differ depending on the type of antigen and the affinity of the selected T cell receptor (TCR), particularly to genetic polymorphisms and especially for *PD-1* and *PDL-1*. However, how *PD-1* or *PDL-1* polymorphisms could influence irAE occurrence remains unknown. Furthermore, regarding the exposure to steroids or other immunosuppressant drugs for the first irAEs, up until now, no data have explored the deletion of latent auto-immune cells. This explains why treatment of the first irAEs does not mean that the immune system has been completely remodeled and that all latent auto-immune cells are destroyed. This raises the following question: Are second-irAE-related cells induced by the first exposure to ICI, the second or both? Of course, translational studies would help to answer these questions. From a clinical perspective, an indirect way to explore this condition would be to consider (i) the complete clearance of the first exposure to ICI, (ii) type of second irAE, (iii) and its time to onset after ICI resumption. If the first ICI is completely cleared and the onset of the second irAE occurs in the same timeframe as the first ICI exposure, then it is reasonable to think that the second irAE is clearly related to the rechallenge (i.e., a new auto-immune thyroid disorder occurring 8 weeks after second exposure to ICI). If the time lapse is shorter, it could be reasonable to believe that the latent auto-immune cells were stimulated by both exposures. If the time lapse is longer and the patient is exposed to steroids or immunosuppressants, it could be either that (1) latent auto-reactive cells expanded when exposed to the first ICI but were modulated by the treatment of the first irAE and restimulated by the second exposure to ICI, or that (2) latent auto-reactive cells expanded during the second exposition to ICI in an immunosuppressive environment. It could also help to distinguish a relapse of the first irAE after rechallenge from the onset of a second irAE with the same phenotype but driven by different autoreactive cells, if the relapse occurs a shorter time after rechallenge than the second irAE. Unfortunately, many irAEs were shown not to occur on the typical timeline after exposure to ICI or to ICI combinations. As a result, it is impossible to know if irAEs occurring after ICI resumption are clearly related to ICI second exposure or not. By default, irAEs occurring after ICI resumption were assumed to be related to the second exposure to ICIs (or combination of both exposures).

Moreover, most publications brought together what they termed resumption, rechallenge and retreatment in their evaluation of the ICI safety profile. Retreatment is the re-administration of the same therapeutic class following tumor relapse. Rechallenge is defined as reintroduction, after an intervening treatment of the same therapy to which the tumor has already proven to be resistant [28]. None of these definitions address the issue of restarting ICI after irAE, given that the treatment has not been completed, and that the tumor is not yet officially resistant to ICI. Resumption as the restarting after an interruption would be the best definition (Figure 1). To differentiate a postponed treatment from a resumption, the threshold of the biological activity of each ICI should be known. The postponed treatment would be defined as a reinjection of ICI while the level of ICI is above the threshold of the biological activity, whereas resumption would occur when the reinjection of ICI is below the threshold of the biological activity. As of now, no clear definition of resumption exists, a factor further complicating the interpretation of the onset of second irAEs. One solution would be to base the definition either on pharmacokinetic parameters (five half-lives), but that would be very restrictive (e.g., the half-life of nivolumab is about 21 days), or on a multiple of the recommended interval between two treatments.

Table 1 summarizes the available data on the timing of ICI resumption after a first irAE, showing that it ranges from a median of 14 days [29] to 60 weeks [30]. It also bears mentioning that some studies have not reported this variable but provided a clear minimal time lapse between ICI discontinuation and ICI resumption. The heterogeneity of the data may lead to the consideration of a second irAE after either postponed or resumed treatment in the same publication. It is impossible to know whether or not the risk of a second irAE is the same in the postponed or in the resumption situation.

In conclusion, the analysis of the available data shows that their heterogeneity may lead to a misinterpretation of the risk of a second irAE. That is why a data threshold of minimal biological activity for ICIs would help us to more precisely determine the factors related to postponed treatment, as opposed to resumption. Moreover, translational studies would help to calculate the percentage of irAEs related to immune stimulation from the first rather than the second ICI exposure.

## 3. Safety of Resumption of ICI after a First irAE

Only studies in English containing data on more than 20 patients with ICI resumption after ICI discontinuation for irAEs were retained. Table 2 summarizes these selected studies. We excluded one study because there were no data on the interrupted ICI and on the sub-group of patients with reinitiated ICI [45]. Finally, a total of 1563 patients from 18 studies received ICI resumption after ICI discontinuation due to irAEs. After the exclusion of the studies without details on the grade of initial irAEs or those including only patients with grade ≥3 irAEs [21,35,38,39], 47% (33.0–55.0) of the remaining patients with ICI resumption had grade ≥3 initial irAEs. As mentioned above, all irAEs occurring after ICI resumption were considered as second irAE.

Even though all of these studies are retrospective, they all present a Newcastle Ottawa Scale score ≥6, reflecting high quality. Most are multicentric, except for four studies [30,39,40,42]. Only four studies contained data of more than 100 patients with ICI resumption [21,32,33,38].

Aside from heterogeneity in the timing range of ICI resumption (Table 1), the studies were heterogeneous in terms of malignancy type and ICI regimen. Moreover, different studies assessed the safety of ICI resumption through the perspective of cancer type (i.e., renal cell carcinoma, melanoma, non-small-cell lung cancer) [31,34,37,39,40,41,42,44,46], irAE type (i.e., acute kidney injury, immune-mediated diarrhea and colitis (IMDC)) [32,35,38] or both [21,29,30,33,36,43].

The recurrence rate of any irAE after ICI resumption was 45.6% (36.5–50.0), of which 37.2% (30.7.56.7) were grade ≥3. The recurrence rate of the same irAE was 22.6% (15.4–26.8), of which 50.0% (41.2–50.0) were grade ≥3 (24/51). After the exclusion of fatal irAEs, irAE recurrence led to permanent ICI discontinuation in 19.4% (10.4–26.9) (100/571) of rechallenged patients.

Very few data are available on the risk factors associated with irAE recurrence. One study showed that the first irAEs occurred earlier in the patients who experienced new or recurrent irAEs after ICI resumption (9 vs. 15 weeks, *p* = 0.04) [43]. Contrary results were observed in another study with a longer duration between ICI discontinuation and ICI resumption (112 vs. 63 days, *p* = 0.01) [33]. Data on organ-specific irAEs are more consistent. For example, patients with initial gastrointestinal irAEs were more likely to have recurrent grade ≥2 irAEs after ICI resumption compared to those with no gastrointestinal irAEs [21,33]. By contrast, initial endocrine disorders were less likely to recur [21,33]. Only one study, which focused on IMDC, reported in a multivariate analysis that the risk factors associated with IMDC recurrence were initial grade ≥3 IMDC (OR 2.19, 95% CI (0.66–7.29), *p* = 0.20), initial need for immunosuppressive therapy (OR 3.22, 95% CI (1.08–9.62), *p* = 0.019) and longer duration of initial IMDC symptoms (OR 1.01, 95% CI (1.00–1.03), *p* = 0.031) [32]. The resumption of ICI combination after initial ICI combination was anecdotal insofar as the aim of resumption is to maintain ICI treatment. Most of the resumption scenarios were to (re)start with anti PD(L)-1 therapy after at least one combination of either anti-CTLA-4- or anti PD(L)-1-related irAE. We recently showed that the rechallenge of the same ICI drug or the same ICI combination was associated with a lower rate of irAE recurrence (77.1% vs. 90%, *p* = 0.02) [33]. In addition, Pollack et al. concluded that patients who experienced colitis or hypophysitis after an anti-PD-1/anti-CTLA-4 combination could safely resume anti-PD-1 [41]. The resumption of anti-CTLA-4 after anti-CTLA-4-related irAE or ICI combination was reported in fewer than 70 patients [21,32,36,39]. Abu-Sbeih et al. reported the recurrence of IMDC in 44% of patients who received anti-CTLA-4 resumption compared to 32% who received anti-PD(L)-1, and reduced risk of IMDC recurrence in the anti-PD(L)-1 group compared to the anti-CTLA-4 resumption group in multivariate analysis (odds ratio (OR) 0.30; 95% CI 0.11–0.81; *p* = 0.019) [32].

Overall, despite a sizable heterogeneity between the studies, these results show that the recurrence rates of irAE remain globally concordant with relatively tight ranges. They highlight (i) that more than half of patients with ICI discontinuation for irAE did not have irAE recurrence after ICI resumption; (ii) that the second irAE is often different from the first irAE [30,31,34,39,41,46]; (iii) that the second irAE seems no more severe than the first irAE; and (iv) the possibility of managing second irAEs while continuing ICI in most patients.

However, these results must be interpreted with caution given the retrospective design of the studies, heterogeneity of the timing of resumption and low sample sizes. All in all, they give an overview of ICI resumption after irAEs.

## 4. Unmet Medical Needs

Several general or organ-specific recommendations have been published to help to manage irAEs [17,47,48,49,50,51,52,53,54,55,56,57,58]. Generally, ICI should be continued with close monitoring for grade 1 irAEs, except for some neurologic, hematologic, and cardiac toxicities. Grade 2/3 irAEs require corticosteroids (0.5–1 or 1–2 mg/kg a day of prednisone equivalent, respectively) and temporary ICI discontinuation with the consideration of resuming when symptoms revert to grade ≤1. If symptoms do not improve within 48–72 h of high-dose steroids, various immunosuppressants (e.g., infliximab, rituximab) may be offered depending on the organ involved. ICI dose adjustments are not recommended. Finally, the permanent discontinuation of ICI is recommended for grade 4 irAEs (except for endocrinopathies if controlled by hormone replacement) [17]. However, given the relatively limited knowledge of irAEs and the lack of robust available data (mostly heterogeneous, retrospective, and small-scale studies), these recommendations are currently based only on expert opinion with a low level of evidence (C). The benefit/risk ratio of resumption for neurologic or cardiac involvements that can be potentially life-threatening is clearly in favor of the discontinuation of the ICIs. For gastro-intestinal toxicities, the balance is unclear due to the high risk of recurrence of irAEs; it is based mostly on the oncologic situation and the patient’s motivation. For the other irAEs, if the oncologic situation favors the resumption of the ICI, we encourage oncologists to discuss with their patients the resumption of ICI with the close management of potential second irAEs.

The heterogeneity of irAEs further complicates their management. In fact, irAEs can involve any organ [8], mostly within 2–16 weeks of ICI initiation [12], but some cases have been reported only a few days after ICI initiation or ≥1 year after the last ICI infusion [10,11]. The severity and type irAEs also vary depending on the ICI regimen. More precisely, grades ≥3 irAEs were more frequent, with anti-CTLA-4 compared to anti-PD-1 (31% vs. 10%) in a systematic review [13]. Combination therapies were associated with a greater risk and up to fivefold shorter median time until irAE compared to monotherapy (32 days vs. 146 days) [14]. Moreover, tumor-specific patterns of irAEs were also reported (i.e., melanoma associated with more dermatological/gastrointestinal irAEs and fewer cases of pneumonitis than other cancers treated by ICI) [37,59], suggesting that different organ-specific microenvironments could drive specific irAEs. Finally, inter-individual variations in the development of irAEs for the same cancer and the same ICI regimen suggest an additional genetic background [15] with a predisposition to autoimmunity (i.e., CTLA-4 and PDL-1 polymorphisms that are associated with autoimmune disorders [60,61]. Whether pre-existing autoimmune disease (AID) is a risk factor for irAEs remains controversial, as these patients were excluded from clinical trials. A meta-analysis of 14 retrospective studies showed a pooled incidence of AID flare and de novo irAEs, whatever the grade, of 35% and 33%, respectively, with no differences between patients with or without immunosuppressive therapy at the start of an ICI [62]. Two more recent retrospective studies of 27 and 47 patients with pre-existing AID treated by an ICI showed an overall AID exacerbation in 26 to 52% of patients, respectively. More than half of these patients required immunosuppressive drugs and 14 to 42% underwent ICI discontinuation [30,63]. Finally, a significant higher risk of irAEs or severe irAEs among patients with pre-existing AID compared to patients without pre-existing AID and treatment by ICI were reported [59,64]. Conversely, van der Kooij et al. reported an overall incidence of grade ≥3 irAEs as being similar in patients with or without preexisting AID, except for severe colitis among patients with preexisting inflammatory bowel disease [65]. These results suggest that irAEs are common in patients with preexisting AID but that they are often mild and manageable, without discontinuing therapy.

Identifying predictive risk factors not only for overall irAE occurrence, but also for irAE recurrence, is a challenging but necessary research perspective. Some clinical factors have been linked to the development of irAEs, such as preexisting AID, the use of CTLA-4 inhibitors and grade ≥3 renal insufficiency [64,66]. Obesity was reported as independently associated with irAEs whatever the grade, grade ≥3 irAEs [67] and ICI discontinuation [67] in patients treated with pembrolizumab [67,68], but this association was not confirmed with atezolizumab [69] or ipilimumab [70]. Additionally, a recent review summarized the available data on biomarkers linked to ICI-related toxicities [71]. Importantly, while these studies assessed the risk factors of irAE occurrence after the first ICI treatment, very few and sometimes conflicting data are available regarding the risk factors of irAE recurrence, as discussed above [32,33,41,43]. In the same way, while the type of irAEs after the first ICI treatment is generally well-described (e.g., colitis, hypophysitis and rash are more frequently described with anti-CTLA-4 and pneumonitis, hypothyroidism, arthralgia and vitiligo with anti-PD-1) [13], data remain unclear and heterogeneous on the type of second irAEs according to the type of ICI resuming. Finally, one of the main limitations to knowledge of ICI resumption is the lack of clear data on overall survival (OS), tumor by tumor, or compared within a group of tumors with homogenous prognosis. A growing body of evidence suggests that irAE occurrence is predictive of anti-PD(L)-1) response with a marked improvement in progression-free survival, OS and overall response rate in patients with irAE compared to those without [72]. Moreover, some specific irAEs are associated with enhanced survival, cutaneous irAEs being one of the best documented examples [73,74,75,76,77]. However, to date, whether or not specific irAE-related factors (severity, timing of onset, therapeutic intervention, irAE recurrence) are associated with increased OS remains unknown and questions the need for ICI resumption after ICI discontinuation due to irAE. The aim of resumption is to offer the patient the best available treatment for their cancer without knowing its impact on OS, whereas it is certain that the quality of life is impacted due to the number of infusions and risk of second irAE(s).

Combinatorial approaches, such as ICIs associated with chimeric antigen receptor-T (CAR-T) cells, are currently being investigated to improve antitumor effects, and to mitigate toxicities. However, only a few trials have been performed and most are proofs of concept with small numbers of included patients [78,79]. Adequate prospective and pharmacovigilance studies are needed to assess the safety of such combinations from a long-term perspective.

## 5. Conclusions

ICI resumption after irAE should always be discussed in a multidisciplinary team meeting in light of the usefulness of rechallenge, patient comorbidities and risk of recurrence of first irAE(s). However, although few data are available, ICI resumption is not recommended in clinical practice in some organ-specific irAEs (i.e., myocarditis and/or the central nervous system), especially if severe. ICI resumption should be regarded cautiously after gastrointestinal irAEs due to a high risk of irAE recurrence. Further prospective observational studies are therefore needed to assess risk factors of irAE recurrence to rapidly identify at-risk patients and modalities of ICI resumption and should be coupled with translational studies to improve knowledge of the physiopathology of irAEs.

## Figures and Tables

**Figure 1 cancers-14-00955-f001:**
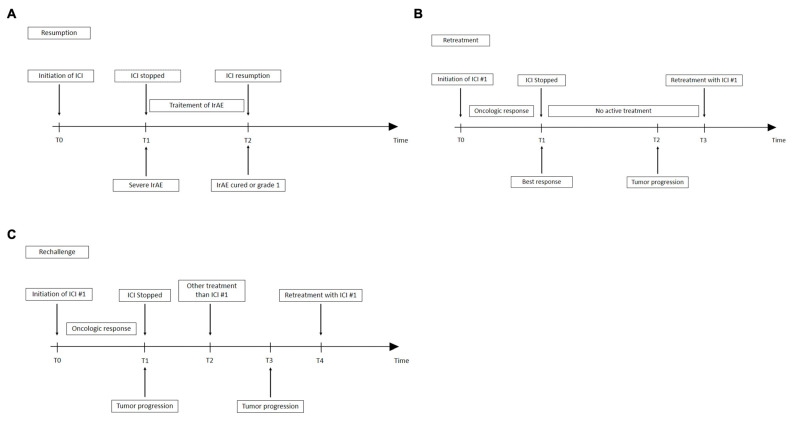
Differences between definitions of resumption (**A**), retreatment (**B**) or rechallenge (**C**) ICI: immune checkpoint inhibitor; irAE: immune-related adverse event; T: time; #1: number 1.

**Table 1 cancers-14-00955-t001:** Summary of available definitions and timing of ICI resumption.

Study	Resumption Definition	Median Time before Resumption
Abou Alaiwi et al. [31]	Dose interruption for at least 1 week due to irAEs	0.9 (range: 0.2–31.6) months
Abu Sbeih et al. [32]	ICI resumption after suspension because of IMDC onset	49 (IQR: 23–136) days
Allouchery et al. [33]	Discontinued ICI before rechallenge for a period at least equal to twice the duration of a cycle	56 (IQR: 42–84) days
Bhatlapenumarthi et al. [30]	Rechallenge after irAE with the same drug and dose	2–30 weeks depending on irAE type and severity
Brunot et al. [34]	Washout between ipilimumab and first dose of anti-PD-1	25 weeks (range: 2–194)
Cortazar et al. [35]	Readministration of the same or different ICI	1.8 months (IQR: 1.2–11.0) after diagnosis of AKI- irAE
de Malet et al. [36]	Second-irAE treatment with ICIs after GI-irAE	1.1 months (range: 0.1–32.6) after the end of GI irAE
Gobbini et al. [37]	At least 12 weeks after discontinuation because of toxicity, disease progression or clinical decision	NR
Gupta et al. [38]	Discontinuation then reinitiation with same, different class or ICI combination	1.9 months (IQR: 1.1–4.0) after initial irAE
Li et al. [39]	ICI resumption after high-grade ICI hepatitis resolution	51 days (IQR: 15–77)
Mouri et al. [40]	Treatment delay of longer than 4 weeks due to an irAE	NR
Patrinely et al. [29]	Initiation of ICI after hepatitis resolution	14 days (median days)
Pollack et al. [41]	Anti-PD-1 resumption after combined anti-CTLA-4 and anti-PD-1	58 days (range: 14–395)
Santini et al. [42]	Treatment delay longer than one week between planned doses of ICI	32 days (range: 7–177)
Simonaggio et al. [43]	Readministration of the same drug class in the same patient	NR
Ravi et al. [44]	At least 2 separate lines of ICI (alone or in combination with other therapies).	NR

AKI: Acute kidney injury; GI: gastro-intestinal; ICI: immunological checkpoint inhibitor(s); IQR (interquartile range); irAE: immune-related adverse event; IMDC: immune-mediated diarrhea and colitis; NR: not reported.

**Table 2 cancers-14-00955-t002:** Recurrence of irAEs after ICI resumption.

Study	NOS Scale (*)	Malignancy	Number of Retreated Patients after 1st irAE	Rechallenge Type	% of Any 2nd irAE (New + Same irAE)	% of Same 2nd irAE	% of Permanent ICI Discontinuation
Abou Alaiwi et al. [31]	8	Metastatic RCC	36 (G ≥3: 47%)	Resumption: Anti-PD(L)1: 67% Combination (33%)	50% (G ≥3: 38.9%)	16.7%	27.8%
Abu Sbeih et al. [32] ^†^	6	Melanoma (54%) NSCLC (16%) Genitourinary (10%) Other solids (16%) Hematologic (4%)	167 (G ≥3 colitis: 33%) ^††^	Resumption: Anti-CTLA-4 (14%) Anti-PD(L)-1 (42.5%) Switch: Anti-CTLA-4 to anti-PD(L)-1 (38.5%) Anti-PD(L)-1 to anti-CTLA-4 (5%)	NR	34.1% (G ≥2 colitis: 46.6%)	NR
Allouchery et al. [33]	7	Melanoma (44%) Lung (41%) RCC (6%) Lymphoma (3%) Others (6%)	180 (G ≥3: 49%)	Resumption: Same ICI or ICI combination (85%) Switch (15%)	38.9% (G ≥3: 35%)	27.2%	26.1%
Bhatlapenumarthi et al. [30]	6	NSCLC (44%) SCLC (11%) Melanoma (18.5%) RCC (18.5%) Anaplastic thyroid (4%) Thymic (4%)	27 (G ≥3: 26%)	Resumption: Anti-PD-1 (93%) Anti-PDL-1 (7%)	33.3% (G ≥3: 55.6%)	7.4%	NR
Brunot et al. [34]	7	Metastatic melanoma	56 (G ≥3: 100%)	Switch: Anti-CTLA-4 to anti-PD-1	35.7% (G ≥3: 60%)	14.3%	8.9%)
Cortazar et al. [35] ^†††^	8	Melanoma (36%) ^¶^ Lung (26%) ^¶^ Genitourinary (17%) ^¶^ Other (21%) ^¶^	31	Resumption with the same ICI (87%)	NR	22.6%	NR
de Malet et al. [36]	7	Melanoma (75%) ^¶^ NSCLC (11%) ^†^ RCC (4%) ^¶^ Prostate (3%) ^¶^ Lymphoma (3%) ^¶^ Cervical (3%) ^¶^ Colorectal (1%) ^¶^ Ovarian (1%) ^¶^	26 (G ≥3: 61.5%)	Resumption: Anti-CTLA-4 (23%) Anti-PD-1 (31%) Switch: Anti-CTLA-4 to anti-PD-1 (42%) Anti-PD-1 to anti-CTLA-4 (4%)	46.2% (G ≥3: 16.7%)	23.1% (G ≥3: 33.3%)	NR
Dolladile et al. [21]	6	No detail about only rechallenged patients	452 (Serious: 92%)	PD(L)-1 (82%) CTLA-4 (5%) Combination (13%)	33.2%	28.8%	4.6% treatment-related death
Gobbini et al. [37]	8	NSCLC	58 (G ≥3: 47%)	Resumption: Anti-PD(L)-1 (100%)	14.8% *	7.4% *	NR
Gupta et al. [38] ^†††^	8	Melanoma (32%) Genitourinary (23%) Lung (22%) Other (22%)	121 (Stage ≥2: 78%)	Resumption: (81%) Switch: (19%)	NR	16.5% (Stage ≥2: 80%)	0%
Li et al. [39]	7	Melanoma	31 (G ≥3: 100%)	Resumption: Anti-PD-1 (84%) Anti-CTLA-4 (6.5%) Anti-PDL-1 (3%) Switch: Anti-CTLA-4 to anti-PD-1 (6.5%)	48.4% (G ≥3: 20%)	12.9% (G ≥3: 50%)	19.3%
Menzies et al. [46]	7	Melanoma	67 (G ≥3: 87%)	Switch: Anti-CTLA-4 to anti-PD-1 (100%)	34.3% (G ≥3: 60.9%)	3%	11.9%
Mouri et al. [40]	8	Stage III/IV NSCLC	21 (G ≥3: 33%)	Resumption: anti-PD-1 (100%)	71.4% (G ≥3: 6.7%)	NR	NR
Pollack et al. [41]	7	Metastatic melanoma	80 (G ≥3: 69%)	Resumption: Anti-PD-1 (100%)	50% (G ≥3: 35%)	17.5% (G ≥3: 50%)	30%
Patrinely et al. [29] ^‡^	8	Melanoma (84%) ^¶^ Lung (7%) ^¶^ RCC (3%) ^¶^ Squamous cell (1.2%) ^¶^ Other (4%)^¶^	66 (G ≥3: 39%)	NR	39.4%	25.8% (G ≥3: 41.2%)	NR
Ravi et al. [44]	6	RCC	69 (G ≥3: 26%)	Switch or resumption (no details on ICI)	44.9% (G ≥3: 35.5%)	NR	NR
Santini et al. [42]	7	NSCLC	38 (G ≥3: 34%)	Resumption: Anti-PD(L)-1 to anti-PDL-1 (63%) Anti-PD(L)-1 + anti-CTLA-4 to anti-PDL-1 (37%)	52.6% (G ≥3: 40%)	26.3% (G ≥3: 60%)	5.2% treatment-related death
Simonaggio et al. [43]	6	Melanoma (28%) Lung (15%) Lymphoma (15%) Colorectal (15%) Other (30%)	40 (G ≥3: 55%)	Resumption: Anti-PD(L)-1 (100%)	55% (G ≥3: 61.9%)	42.5%	NR

* Among the 27 patients with grade ≥3 irAEs (no data available on the 31 patients with grade 1–2 irAEs); ^†^ Focus only on ICI-related IMBC; ^††^ Among the 138 patients with documented colitis; ^†††^ Focus only on ICI-related AKI; ^¶^ Among patients with overall irAE (no data available on the irAE rechallenged sub-group); ^‡^ Focus only on ICI-related hepatitis; AKI: Acute kidney injury; CTLA-4: cytotoxic T-lymphocyte antigen 4; irAE: immune-related adverse event; G: grade; NSCLC: non-small-cell lung cancer; IMDC: immune-mediated diarrhea and colitis; NOS: Newcastle Ottawa Scale; PD(L)-1: programmed cell death protein(ligand)- 1; NR: not reported; RCC: renal cell cancer; SCLC: small-cell lung cancer.

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
