# Peer review of "Safety of Immune Checkpoint Inhibitor Resumption after Interruption for Immune-Related Adverse Events, a Narrative Review"

_cancers, 2022, doi:10.3390/cancers14040955_

Round 1
Reviewer 1 Report
Considering the current paradigm shift to immuno-oncology, the review topic is timely and the authors have written an excellent review article.
In line 91, authors have hypothesized that ICI mediated AEs can occur well before the initiation of treatment. The rationale behind this hypothesis can be explained little more.
L106, did the authors mean- …second irAE occurs at same time-frame as the first ICI exposure…..
Please define the abbreviations used, for example NOS at L168 etc
Author Response
We would like to thank the reviewers for their comments which will improve the quality of our manuscript. We would also thank the Editor for his work in this process.
Please find below our responses to the reviewers (their questions are in bold to contrast with our answers, the original text is in black, the new text is in blue).
Reviewer 1
- Considering the current paradigm shift to immuno-oncology, the review topic is timely and the authors have written an excellent review article.
We thank the reviewer for this compliment.
- In line 91, authors have hypothesized that ICI mediated AEs can occur well before the initiation of treatment. The rationale behind this hypothesis can be explained little more.
This comment is very interesting and will improve the understanding of our manuscript.
We hypothesized that the break of tolerance happened before starting Immune Checkpoint Inhibitor (ICI) treatment. More precisely, autoreactive cells are already present but probably anergic, without clinical symptoms of auto-immune disease. By releasing the inhibitory brakes of T cells and by activating other cells of the innate and adaptive immune system, ICI could lead to activation of anergic and potential autoreactive cells, which could proliferate and recruit other cells of the immune system to target the auto-epitope, leading to potential development of autoimmune diseases.
The manuscript has been modified (lines 91-99).
“Due to the rapid onset of irAEs (a few weeks after the beginning of the treatment), it could be hypothesized that the break of tolerance at a cellular level occurred before ICI treatment, but without any clinical symptoms of auto-immune disease. As a result, the stereotypical kinetics of the first irAEs (skin irAEs between 3-7 weeks, pulmonary irAEs between 10-16 weeks, hepatitis between 6-15 weeks, colitis between 4-10 weeks, endocrinopathies after 6 weeks for anti-PD-(L)1) [27] may be related to activation and amplification of the preexisting autoimmune cells, which may differ due to the type of antigen, the affinity of the selected T-Cell Receptor (TCR), and probably also to genetic polymorphisms, especially for PD-1 and PDL-1.”
- L106, did the authors mean- …second irAE occurs at same time-frame as the first ICI exposure…
This point has been modified according to the recommendations of the reviewer (line 109).
“If the first ICI has been completely cleared and the onset of the second irAE occurs in the same timeframe as the first ICI exposure, then it is reasonable to think that the second irAE is clearly related to the rechallenge (i.e., a new auto-immune thyroid disorder occurring 8 weeks after second exposure to ICI).”
- Please define the abbreviations used, for example NOS at L168 etc
We would like to apologize for this oversight.
The manuscript has been modified: “Even though all of these studies are retrospective, they all present a “Newcastle Ottawa Scale score ≥6 reflecting high quality.”
We did not find any other undefined abbreviations.

Reviewer 2 Report
In this review, the authors discussed and described the effect of the immune checkpoint inhibitors (ICI) as a monotherapy, and considered it beneficial more than the conventional treatment. Furthermore, they checked the effect of immune-related adverse events (irAEs) after and before treatment of ICI. In this paper, they cited references of a small number of patients with heterogeneous data and they find out the recurrence rate of the same irAE after ICI continuation. However, it still has some questions that need to be further described in the manuscript.
Questions:
Q#1
What could be the possible genetic assessment of the risk factors of irAE recurrence to identify the patients' at-risk? Is any research has been published before?
Q#2
Can authors group or organize the homogenous of irAE contribution toward recurrence and second-line treatment failure to improve the modalities and efficacy of ICI resumption?
Q#3
Can authors discuss the CAR-T cell therapy to improve the cross-reactivity of the immune system to the normal cells to control the irAE complications and recurrence in the discussion of this manuscript?
Q#4
Can we use additional hormone repletion strategies to reduce the combination therapies associated risks to irAE than monotherapy, comprehend the complexities, and improve retreatment decisions? Authors need to further describe in the manuscript.
Q#5
Liver toxicity of irAE has not been discussed.
Author Response
We want to thank the reviewers for their insightful comments
On behalf of all co-authors,
Dr Marion Allouchery
Please find below our responses to the reviewers (their questions are in bold to contrast with our answers, the original text is in black, the new text is in blue).
Reviewer 2
- What could be the possible genetic assessment of the risk factors of irAE recurrence to identify the patients' at-risk? Is any research has been published before?
We would like to thank the reviewer for this excellent question.
Our literature review did not find any study exploring genetic polymorphisms associated with immune-related adverse events (irAEs). As part of the reviewer comment, a new search of PubMed did not identify any study on this topic.
The manuscript has been modified (lines 98, 99).
“However, how PD-1 or PDL-1 polymorphisms could influence irAE occurrence remains unknown. Furthermore, as regards exposure to steroids or other immunosuppressant drugs for the first irAEs, up until now, no data have explored the deletion of latent auto-immune cells.”
- Can authors group or organize the homogenous of irAE contribution toward recurrence and second-line treatment failure to improve the modalities and efficacy of ICI resumption?
Unfortunately, as most studies do not provide individual data on ICIs efficacy according to irAEs, we cannot answer this interesting question.
- Can authors discuss the CAR-T cell therapy to improve the cross-reactivity of the immune system to the normal cells to control the irAE complications and recurrence in the discussion of this manuscript?
As suggested by the reviewer, combinatorial approaches, such as ICIs associated with Chimeric Antigen Receptor-T (CAR-T) cells, are currently being investigated to improve antitumor effects, and to mitigate toxicities. As most of the current data are based on case-reports in multiple myeloma, with ICI treatment after failure of CAR-T cells, we cannot conclude on this point.
However, the manuscript has been modified to address this issue.
“Combinatorial approaches, such as ICIs associated with Chimeric Antigen Receptor-T (CAR-T) cells, are currently being investigated to improve antitumor effects, and to mitigate toxicities. However, only a few trials have been performed and most are proofs of concept with small numbers of patients included [78,79]. Adequate prospective and pharmacovigilance studies are needed to assess the safety of such combinations in a long-term perspective.” (lines 311-316)
- Can we use additional hormone repletion strategies to reduce the combination therapies associated risks to irAE than monotherapy, comprehend the complexities, and improve retreatment decisions? Authors need to further describe in the manuscript.
This remark is very interesting. However, our review focused on irAEs occurrence after ICI resumption. How hormone replacement could influence the onset if second irAEs remains unknown. The male sex has been associated with higher survival rates with ICIs combination in advanced melanoma (Jang et al, 2021). However, to our knowledge, a potential interaction between sex and irAEs has never been studied.
- Liver toxicity of irAE has not been discussed.
The manuscript has been modified (lines 93-99).
“As a result, the stereotypical kinetics of the first irAEs (skin irAEs between 3-7 weeks, pulmonary irAEs between 10-16 weeks, hepatitis between 6-15 weeks, colitis between 4-10 weeks, endocrinopathies after 6 weeks for anti-PD-(L)1) [27] may be related to activation and amplification of the preexisting autoimmune cells, which may differ due to the type of antigen, the affinity of the selected T-Cell Receptor (TCR), and probably also to genetic polymorphisms, especially for PD-1 and PDL-1.”

Reviewer 3 Report
The review article is well written and described.
I just have few minor comments.
1. In Fig. 1, T0, T1 and so on is not clear. Please explain what does "T" stands for.
2. Some of the abbreviations are not explained in the main text but rather in the tables such as NOS in line 168, which is confusing. Please go through the manuscript and explain abbreviations in the main text.
Please review the English and grammar as well.
Author Response
We want to thank the reviewers for their insightful comments
On behalf of all co-authors,
Dr Marion Allouchery
Please find below our responses to the reviewers (their questions are in bold to contrast with our answers, the original text is in black, the new text is in blue).
Reviewer 3
The review article is well written and described.
I just have few minor comments.
- In Fig. 1, T0, T1 and so on is not clear. Please explain what does "T" stands for.
We thank the reviewer for this comment, the meaning was indeed unclear.
“T” means Time. We added this precision in the footnote of figure 1.
- Some of the abbreviations are not explained in the main text but rather in the tables such as NOS in line 168, which is confusing. Please go through the manuscript and explain abbreviations in the main text.
We would like to apologize for this oversight.
The manuscript has been modified: “Even though all of these studies are retrospective, they all present a Newcastle Ottawa Scale score ≥6 reflecting high quality.”
We did not find any other undefined abbreviations.
- Please review the English and grammar as well.
We submitted our modified manuscript to a English-native medical editor , who corrected all the grammar and English.

Reviewer 4 Report
The review entitled “Safety of Immune Checkpoint Inhibitor Resumption after Interruption for Immune-Related Adverse Events, a Narrative Review” by Allouchery et al. is very compact. The topic discussed is important in the context of the safety of ICI therapies, especially in the case of induced in consequence of the treatment adverse events.
However, to fill in the topic that is being touched, according to the authors' notion that IrAE severity and type varies depending on ICI regimen, the table presenting types of irAEs induced in response to ICI treatment as well the listing if the listed irAEs are specific only for a specific ICI or similar for different ones could be included in the article.
It would be also interesting to discuss what irAEs could be induced when immune checkpoint inhibitor resumption would be done after other immunotherapies (e.g., TILs, CAR-T, immune vaccines)? If such an approach could be safer for the patient regarding treatment evoked adverse events?
Moreover, the authors point out about very little data exists about factors associated with the risk or irAE recurrence, it is an important issue that could be discussed in more detail, for instance by proposing possible strategies to prevent and manage irAEs.
Author Response
We want to thank the reviewers for their insightful comments
On behalf of all co-authors,
Dr Marion Allouchery
Please find below our responses to the reviewers (their questions are in bold to contrast with our answers, the original text is in black, the new text is in blue).
Reviewer 4
- The review entitled “Safety of Immune Checkpoint Inhibitor Resumption after Interruption for Immune-Related Adverse Events, a Narrative Review” by Allouchery et al. is very compact. The topic discussed is important in the context of the safety of ICI therapies, especially in the case of induced in consequence of the treatment adverse events.
We thank the reviewer for this comment.
- The table, presenting types of irAEs induced in response to ICI treatment as well the listing if the listed irAEs are specific only for a specific ICI or similar for different ones, could be included in the article.
We would like to thank the reviewer for this interesting comment. First, as the aim of our review was to assess the safety of ICI resumption after a first irAE, we only focused on the frequency and severity of second irAEs. Second, while the type of irAEs after first ICI treatment is actually well-described, data remain unclear and heterogeneous on the type of second irAEs according to the type of ICI resuming. Unfortunately, most studies do not provide details for each patient on the type of second irAEs occurring after ICI resumption. As a result, we cannot provide a specific table on this point.
We added a sentence in the text to explain the lack of data about potential association between second irAE and specific ICI: “In the same way, while the type of irAEs after first ICI treatment is generally well-described (e.g., colitis, hypophysitis and rash are more frequently described with anti-CTLA-4 and pneumonitis, hypothyroidism, arthralgia and vitiligo with anti-PD-1) [13], data on the type of second irAEs according to the type of ICI resuming remain unclear and heterogeneous.” (lines 294-298).
- It would be also interesting to discuss what irAEs could be induced when immune checkpoint inhibitor resumption would be done after other immunotherapies (e.g., TILs, CAR-T, immune vaccines)? If such an approach could be safer for the patient regarding treatment evoked adverse events?
We fully agree with the reviewer on this interesting point. This question is challenging: Is it possible to combine therapies (e.g., ICI plus CAR-T cells) to improve the efficacy of old or new adoptive immunotherapies? However, we have not found any data about the efficacy and toxicity of such combinations in a long-term perspective (more than 6 months). We added the small paragraph below at the end of the “Unmet medical needs” part (lines 307 to 313) with 2 additional references [78, 79] (Kverneland et al, J. Immunother. Cancer 2021 and Kverneland et al, Oncotarget 2020):
“Combinatorial approaches, such as ICIs associated with Chimeric Antigen Receptor-T (CAR-T) cells, are currently being investigated to improve antitumor effects, and to mitigate toxicities. However, only a few trials have been performed and most are proofs of concept with small numbers of patients included [78,79]. Adequate prospective and pharmacovigilance studies are needed to assess the safety of such combinations in a long-term perspective.” (lines 311-316)
- Moreover, the authors point out about very little data exists about factors associated with the risk or irAE recurrence, it is an important issue that could be discussed in more detail, for instance by proposing possible strategies to prevent and manage irAEs.
Indeed, as suggested by the reviewer, defining the risk factors of irAEs recurrence is essential to optimize ICI treatment. However, no risk factor of irAE recurrence has been clearly identified in previous studies. We have already discussed this point in the “Safety of Resumption of ICI after a first irAE” part (lines 182 to 206), and that is why we did not provide details in the “Unmet medical needs” part. We returned to this part with the modified sentence: “Importantly, while these studies assessed factors linked to the risk of irAE occurrence after first ICI treatment, very few and sometimes conflicting data are available about risk factors associated of irAE recurrence as discussed above [32,33,41,43]” (lines 290-293). Moreover, prospective studies with high quality data collection are needed to better identify risk factors of irAE recurrence. Concerning the management of irAEs, national and international recommendations based on expert opinions are available. They are summarized in the first paragraph of the “Unmet medical needs” part (lines 234 to 243). In our opinion, data are presently too scarce to propose standardized management for irAEs after ICI resumption. In our experience, second irAEs should be managed in the same manner as we did for first irAEs.

Round 2
Reviewer 4 Report
I endorse the publication of the manuscript in its current form.